Synthetic surfactant containing SP-B and SP-C mimics is superior to single-peptide formulations in rabbits with chemical acute lung injury

Walther Frans J. 1 2 fjwalther@ucla.edu
Hernández-Juviel José M. 1
Gordon Larry M. 1
Waring Alan J. 3 4 5
1 Department of Pediatrics, Division of Medical Genetics, Los Angeles Biomedical Research Institute, Harbor-UCLA Medical Center , Torrance, CA , USA
2 Department of Pediatrics, David Geffen School of Medicine, University of California Los Angeles , USA
3 Department of Medicine, Division of Molecular Medicine, Los Angeles Biomedical Research Institute, Harbor-UCLA Medical Center, Harbor-UCLA Medical Center , Torrance, CA , USA
4 Department of Medicine, David Geffen School of Medicine, University of California Los Angeles , USA
5 Department of Physiology & Biophysics, School of Medicine, University of California Irvine , CA , USA
Palomo Jose
Electronic publication date: 2014 May 22
Publication date: 2014
Volume: 2
Electronic Location ID: e393
Received 2014 Mar 17; Accepted 2014 May 2
Copyright: © 2014 Walther et al.
Copyright year: 2014
Copyright holder: Walther et al.
License: This is an open access article distributed under the terms of the Creative Commons Attribution License, which permits unrestricted use, distribution, reproduction and adaptation in any medium and for any purpose provided that it is properly attributed. For attribution, the original author(s), title, publication source (PeerJ) and either DOI or URL of the article must be cited.
License URL: https://creativecommons.org/licenses/by/4.0/

Keywords: Synthetic surfactant, Hydrochloric acid, Oxygenation, Ventilated rabbits, Surfactant protein B, Lung lavage, Acute lung injury, Surfactant protein C, Lung compliance, Captive bubble surfactometry

Funding: National Institutes of Health R01ES015330 The authors received financial support from the National Institutes of Health through grant R01ES015330. The funder had no role in the design and conduct of the study, in the collection, analysis, and interpretation of the data, and in the preparation, review, or approval of the manuscript.

==============================
Background. Chemical spills are on the rise and inhalation of toxic chemicals may induce chemical acute lung injury (ALI)/acute respiratory distress syndrome (ARDS). Although the pathophysiology of ALI/ARDS is well understood, the absence of specific antidotes has limited the effectiveness of therapeutic interventions.

Objectives. Surfactant inactivation and formation of free radicals are important pathways in (chemical) ALI. We tested the potential of lipid mixtures with advanced surfactant protein B and C (SP-B and C) mimics to improve oxygenation and lung compliance in rabbits with lavage- and chemical-induced ALI/ARDS.

Methods. Ventilated young adult rabbits underwent repeated saline lung lavages or underwent intratracheal instillation of hydrochloric acid to induce ALI/ARDS. After establishment of respiratory failure rabbits were treated with a single intratracheal dose of 100 mg/kg of synthetic surfactant composed of 3% Super Mini-B (S-MB), a SP-B mimic, and/or SP-C33 UCLA, a SP-C mimic, in a lipid mixture (DPPC:POPC:POPG 5:3:2 by weight), the clinical surfactant Infasurf®, a bovine lung lavage extract with SP-B and C, or synthetic lipids alone. End-points consisted of arterial oxygenation, dynamic lung compliance, and protein and lipid content in bronchoalveolar lavage fluid. Potential mechanism of surfactant action for S-MB and SP-C33 UCLA were investigated with captive bubble surfactometry (CBS) assays.

Results. All three surfactant peptide/lipid mixtures and Infasurf equally lowered the minimum surface tension on CBS, and also improved oxygenation and lung compliance. In both animal models, the two-peptide synthetic surfactant with S-MB and SP-C33 UCLA led to better arterial oxygenation and lung compliance than single peptide synthetic surfactants and Infasurf. Synthetic surfactants and Infasurf improved lung function further in lavage- than in chemical-induced respiratory failure, with the difference probably due to greater capillary-alveolar protein leakage and surfactant dysfunction after HCl instillation than following lung lavage. At the end of the duration of the experiments, synthetic surfactants provided more clinical stability in ALI/ARDS than Infasurf, and the protein content of bronchoalveolar lavage fluid was lowest for the two-peptide synthetic surfactant with S-MB and SP-C33 UCLA.

Conclusion. Advanced synthetic surfactant with robust SP-B and SP-C mimics is better equipped to tackle surfactant inactivation in chemical ALI than synthetic surfactant with only a single surfactant peptide or animal-derived surfactant.

Introduction

Acute lung injury (ALI) and acute respiratory distress syndrome (ARDS) are characterized by severe hypoxic respiratory failure and poor lung compliance, mostly caused by lung injury due to pneumonia, aspiration, sepsis and trauma (Bernard et al., 1994; Matthay, Ware & Zimmerman, 2012). Despite advances in respiratory support, morbidity and mortality of ALI/ARDS continue to be elevated due to a lack of efficient therapeutic modalities. Surfactant replacement therapy has long been considered to be a logical pharmacologic approach as the poor lung compliance in ALI/ARDS is associated with surfactant deficiency and inactivation. However, whereas animal-derived surfactant preparations are highly active in preventing and treating respiratory distress syndrome (RDS) in premature infants (Polin & Carlo, 2014), their efficacy in pediatric and adult patients with ALI/ARDS has been utterly disappointing (Willson et al., 2005; Czaja, 2007). After recombinant SP-C based surfactant (Venticute®, Nycomed GmbH, Konstanz, Germany) failed to improve oxygenation in a recent randomized clinical trial because of insufficient surface activity (Spragg et al., 2004; Spragg et al., 2011), doubts have risen about the rationale for exogenous surfactant treatment in ALI/ARDS (Brower & Fessler, 2011; Dushianthan et al., 2012).

Chemical spills are frequently in the news and the lung has come under attack from chemical spills and even bioterrorism (Maynard & Tetley, 2004; White & Martin, 2010). The lung’s response to inhalation injury of toxic chemicals (chemical ALI) ranges from reactive airways dysfunction syndrome (RADS) to ALI/ARDS with possible fatal outcome (Maynard & Tetley, 2004; Gorguner et al., 2004; White & Martin, 2010). Depending on the toxicant, dose and host factors, cell damage (necrosis or apoptosis) will erode the capillary-alveolar barrier and increase the “leakage” of plasma proteins and water into the interstitium and ultimately into the alveolar spaces. Proteinaceous alveolar edema inactivates lung surfactant and critically affects lung function by reducing lung compliance. Reaction of toxicants with double bonds in unsaturated lipids in cell membranes and lung surface fluids can start a cascade of free radical reactions that further damages cells. Peroxides and carbonyls produced in these reactions may elicit inflammation in the lung via the prostaglandin and leukotriene cascades and magnify and disseminate the toxic injury. The debris of damaged cells and a wide variety of mediators, such as cytokines and chemokines, released by airway epithelium, will attract macrophages and other lung matrix cells and activate various cell surface adhesion molecules, i.e., integrins, on the vascular and respiratory surface. Although the pathophysiology of toxic inhalation injury is well understood (Matthay, Ware & Zimmerman, 2012), specific antidotes to toxic inhalants are not yet available. A potential approach towards chemical ALI, and possibly ALI/ARDS in general, may be offered by the use of a new generation of synthetic surfactant that effectively counteracts surfactant inactivation due to vascular leakage of serum proteins, acute inflammation and oxidative stress (Walther et al., 2000; Walther et al., 2007; Walther et al., 2010; Curstedt, Calkovska & Johansson, 2013; Johansson et al., 2003).

Native lung surfactant is a complex mixture that plays a pivotal role in normal breathing because of its ability to reduce alveolar surface tension to low values and consists of ∼80% phospholipids, 10% neutral lipids and 10% proteins (Notter, 2000). Its biophysical activity depends uniquely on the hydrophobic surfactant proteins B (SP-B) and C (SP-C) (Goerke, 1998; Johansson, 1998; Whitsett & Weaver, 2002). SP-B is a 79 amino acid, lipid-associating monomer (MW ∼8.7 kDa) in humans that is found in the lung as a covalently linked homodimer. Each SP-B monomer consists of 4–5 α-helices with three intramolecular disulfide bridges (i.e., Cys-8 to Cys-77, Cys-11 to Cys-71 and Cys-35 to Cys-46) (Johansson, Curstedt & Jornvall, 1991), and belongs to the saposin protein superfamily. Surfactant protein C (SP-C) is a short (35 amino acids; MW of 4.2 kDa in humans) protein that is highly enriched in valine, leucine and isoleucine residues, making it much smaller and more hydrophobic than SP-B. The sequence of porcine SP-C (i.e., NH2-RIPCCPVNLKRLLVVVVVVVLVVVVIVGALLMGL-COOH), which is highly homologous to that of human SP-C, has an N-terminus with a pair of vicinal Cys residues linked to palmitoyl moieties via thioester bonds, thereby producing a true “proteolipid” (Curstedt et al., 1990). These palmitoyl groups are adjacent to a short polar, N-terminal segment characterized by cationic residues such as lysine and arginine. The N-terminal region of SP-C (residues 1–8) is followed by an extremely hydrophobic polyvaline sequence with residues 9 to 34 forming a stable α-helix in the mid- and C-terminal regions (PDB accession code: 1SPF). When incorporated into lipids, Fourier transform infrared (FTIR) spectroscopy also showed that dipalmitoylated SP-C is principally α-helical, with its long molecular helix axis parallel to the phospholipid acyl chains (Pastrana, Maulone & Mendelsohn, 1991; Vandenbussche et al., 1992). Additional results indicated that high surfactant activities for SP-C peptides were closely correlated with enhanced α-helicity (Johansson et al., 1995; Wang et al., 1996; Johansson, 1998).

Recent work by our group on synthetic SP-B peptides and that of Johansson and Curstedt (Karolinska Institute, Stockholm, Sweden) on synthetic SP-C peptides has led to the creation of highly surface active peptide mimics of SP-B and SP-C, i.e., Super Mini-B (S-MB) (Walther et al., 2010) and SP-C33 (Johansson et al., 2003). S-MB is a 41-residue SP-B mimic (primary sequence NH2-FPIPLPYCWLCRALIKRIQAMIPKGGRMLPQLVCRLVLRCS-COOH) that reproduces the topology of the N- and C-terminal domains of SP-B by joining the N-terminal (residues 1–25) and C-terminal (residues 63–78) α-helices with a custom β-turn that replaces SP-B residues 26–62. Thus, S-MB is a ‘short-cut’ version of SP-B, in which the neighboring N- and C-terminal α-helices adopt a helix-turn-helix motif that is cross-linked by disulfide bridges at Cys-8 to Cys-40 and Cys-11 to Cys-34. S-MB has been shown to be highly surface active in vitro and in vivo, and this may be partially due to its positively charged amphipathic helices binding to anionic surfactant lipids (Waring et al., 2005; Walther et al., 2010). The function of SP-C is highly dependent on preservation of its transmembrane α-helix in lipids and targeted amino-acid replacements have enabled the development of SP-C33, an SP-C mimic with enhanced α-helicity in lipids and surface activities, resembling those of native SP-C (Johansson et al., 2003; Almlén et al., 2010). Specifically, the 33-residue SP-C33 (primary sequence, NH2-IPSSPVHLKRLKLLLLLLLLILLLILGALLMGL-COOH) was adapted from the native porcine sequence (see above), in which the vicinal palmitoylcysteines were replaced by serines, ten valine residues were swapped with leucines and a cationic amino acid was moved from the N-terminus closer to the other positively charged residues (Johansson et al., 2003; Almlén et al., 2010). Surface activity is not only dependent on the quantity and quality of surfactant peptides in lipid mixtures, but also on the lipid constituents themselves, their interaction with surfactant peptides, and the viscosity of peptide/lipid mixtures (Tanaka et al., 1986; Walther et al., 2005; Seurynck-Servoss et al., 2007). Recent research has led us to formulate advanced SP-B and SP-C peptides in a lipid mixture that mimics the composition of native lung surfactant (Notter, 2000; Walther et al., 2005).

Although in vitro methods can rapidly provide information about surface activity of experimental surfactant preparations, whole-animal studies are still necessary to test potential treatment modalities in chemical lung injury. Surfactant deficiency induced by repeated saline lung lavages (Ito et al., 1996; Walther et al., 1998) and surfactant dysfunction induced by intratracheal instillation of hydrochloric acid (Chiumello, Pristine & Slutsky, 1999; Brackenbury et al., 2001) in rats and rabbits are established animal models for ALI/ARDS. Both models allow serial measures of arterial blood gases and represent a relatively pure state (over the first 6–9 h or so) of surfactant dysfunction in animals with mature lungs. In this study, we tested the potential of advanced synthetic surfactant preparations to stabilize and improve lung function in adult rabbits with ALI/ARDS induced by lung lavages and chemical exposure.

Materials and Methods

Materials

Peptide synthesis reagents were purchased from Applied Biosystems (Foster City, CA), high performance liquid chromatography (HPLC) solvents from Fisher Chemical Co. (Pittsburgh, PA), and all other chemicals from Sigma Chemical Co. (St. Louis, MO) and Aldrich Chemical Co. (Milwaukee, WI). Dipalmitoylphosphatidylcholine (DPPC), palmitoyloleoyl-phosphatidylcholine (POPC) and palmitoyloleoylphosphatidylglycerol (POPG) were from Avanti Polar Lipids (Alabaster, AL). The clinical surfactant Infasurf® (Calfactant), a bovine lung lavage extract, was a generous gift of Ony Inc. (Amherst, NY). Young adult New Zealand White rabbits, weighing 1.0–1.3 kg, were obtained from I.F.P.S. (Norco, CA).

Synthesis and characterization of surfactant peptides

S-MB peptide (Walther et al., 2010) and SP-C33 (Johansson et al., 2003) were synthesized on a Symphony Multiple Peptide Synthesizer (Protein Technologies, Tucson, AZ) with standard FMoc chemistry (Walther et al., 2010). Crude peptides were purified by reverse phase HPLC, molecular weights of the peptides were verified by MALDI-TOF, and α-helicity was determined by FTIR spectroscopy. Disulfide connectivities for S-MB (i.e., Cys-8 to Cys-40 and Cys-11 to Cys-34) were confirmed by mass spectroscopy of enzyme-digested fragments (trypsin and chymotrypsin digestion) (Walther et al., 2010). We renamed the SP-C33 analog produced in our lab as SP-C33 UCLA to distinguish it from SP-C33 made by Chiesi Pharmaceutici SpA (Parma, Italy).

Surfactant preparations

Synthetic surfactant preparations were formulated by mixing synthetic phospholipids, consisting of 5:3:2 (weight ratio) DPPC:POPC:POPG, with 3% S-MB, 3% SP-C33 UCLA, or 1.5% S-MB + 1.5% SP-C33 UCLA. All surfactant preparations were formulated at a concentration of 35 mg phospholipids/ml. The composition of the synthetic phospholipid mixture was based on the lipid composition of native lung surfactant (Notter, 2000; Walther et al., 2005). Infasurf, which contains 35 mg/ml of phospholipids with 1.5% proteins, of which 0.8% is SP-B, was used as positive control and synthetic lipids alone as negative control.

Captive bubble surfactometry

Adsorption and dynamic surface tension lowering ability of all surfactant preparations were measured with a captive bubble surfactometer at physiological cycling rate, area compression, temperature, and humidity (Walther et al., 2010). We routinely analyze surfactant samples of 1 µl (35 mg phospholipids/ml) in the captive bubble surfactometer and perform all measurements in quadruplicate.

Animal studies

The animal studies were reviewed and approved by the Institutional Animal Care and Use Committee of the Los Angeles Biomedical Research Institute at Harbor-UCLA Medical Center (Research Project # 12507). All procedures and anesthesia were in accordance with the American Veterinary Medical Association (AMVA) Guidelines.

Young adult New Zealand white rabbits (weight 1.0–1.3 kg) received anesthesia with 50 mg/kg of ketamine and 5 mg/kg of acepromazine intramuscularly prior to placement of a venous line via a marginal ear vein. After intravenous administration of 1 mg/kg of diazepam and 0.2 mg/kg of propofol, a small incision was made in the skin of the anterior neck for placement of an endotracheal tube and a carotid arterial line. After placement of the endotracheal tube, muscle paralysis was induced with intravenous pancuronium (0.1 mg/kg). During the ensuing duration of mechanical ventilation, anesthesia was maintained by continuous intravenous administration of 3 mg/kg/h of propofol and intravenous dosages of 1 mg/kg of diazepam as needed; muscle paralysis was maintained by hourly intravenous administration of 0.1 mg/kg of pancuronium. Heart rate, arterial blood pressures and rectal temperature were monitored continuously (Labchart® Pro, ADInstruments Inc., Colorado Springs, CO, USA). Respiratory function was followed by measurements of arterial pH and blood gases and dynamic lung compliance at 15 min intervals. Dynamic lung compliance was calculated by dividing tidal volume/kg body weight by changes in airway pressure (peak inspiratory pressure minus positive end-expiratory pressure) (ml/kg/cm H2O). Maintenance fluid was provided by a continuous infusion of Lactated Ringer’s solution at a rate of 10 ml/kg/h.

After stabilization on the ventilator, lung injury was induced by saline lung lavage or intratracheal administration of hydrogen chloride (HCl). Lung lavaging results in loss of active surfactant, whereas HCl instillation leads to epithelial and endothelial damage, lung hemorrhages and copious edema formation resulting in surfactant dysfunction. When the partial pressure of oxygen in arterial blood (PaO2) was >500 torr at a peak inspiratory pressure <15 cm H2O in 100% oxygen (FiO2 = 1.0), the rabbits underwent repeated saline lung lavages until the PaO2 dropped below 100 torr (average 3 lavages of 30 ml of normal saline, temperature 37 °C) or received intratracheal HCl until the PaO2 dropped below 200 torr (average 2 doses of 1.5 ml/kg of 0.1 N HCl, pH 1.0, at 15 min intervals). Edema fluid appearing in the trachea was removed by suctioning. When the PaO2 was stable at <100 torr in lavaged animals (n = 39) or the PaO2/FiO2 ratio had reached stable values <200 torr (PaO2/FiO2 <40% of pretreatment values) within 30 min in HCl-exposed animals (n = 40), an experimental or positive (Infasurf) or negative (synthetic lipids alone) control surfactant mixture was instilled into the trachea at a dose of 100 mg/kg body weight and a concentration of 35 mg/ml. Group size was 7–8 in the lavaged and 8 in the HCl-exposed rabbits. All rabbits were ventilated using a Harvard volume-controlled animal ventilator (tidal volume 7.5 ml/kg, positive end-expiratory pressure of 3 cm H2O, inspiratory/expiratory ratio of 1:2, 100% oxygen, and a respiratory rate to maintain the PaCO2 at ∼40 mmHg). Airway flow and pressures and tidal volume were monitored continuously with a pneumotachograph connected to the endotracheal tube and a pneumotach system (Hans Rudolph Inc., Kansas City, MO, USA). Animals were sacrificed 2 h after surfactant administration with an overdose of pentobarbital. End-points were gas exchange (arterial pH, PaCO2 and PaO2), pulmonary mechanics (dynamic lung compliance), and bronchoalveolar lavage fluid proteins and lipids.

Protein and lipid measurements of bronchoalveolar lavage fluid (BALF)

Protein and lipid measurements of bronchoalveolar lavage fluid (BALF) collected during the first lung lavage with 30 ml of normal saline to induce surfactant deficiency (lavaged rabbits only) and the first postmortem lavage (both lavaged and HCl-treated rabbits) are shown in Table 1. Protein was measured using the Lowry assay with human albumin as a standard. Phospholipids were measured by extracting BALF samples in chloroform:methanol 2:1 v:v (1 ml of BALF + 4 ml chloroform:methanol), applying the extract to the Fourier transform infrared-attenuated total reflection (FTIR-ATR) plate and drying it before taking a spectrum (Goormaghtigh, Cabiaux & Ruysschaert, 1990).

Table 1 Protein and phospholipid values in bronchoalveolar lavage fluid (BALF).

Protein and phospholipid values (µg/ml) in BALF obtained during the first lavage to induce surfactant deficiency and the first postmortem lung lavage in lavaged and HCl-treated (postmortem BALF only) rabbits. Data are shown as mean ± SEM.

Surfactant	Protein (µg/ml) ± SEM	Phospholipids (µg/ml) ± SEM	
	1st BALF	Postmortem BALF	1st BALF	Postmortem BALF	
Lung lavaged rabbits (n = 7–8 per group)	
S-MB + SP-C33 UCLA	268 ± 12	2,124 ± 130	10.1 ± 2.7	100.5 ± 17.8	
S-MB	302 ± 21	2,358 ± 65	10.8 ± 2.8	96.3 ± 33.0	
SP-C33 UCLA	297 ± 27	2,503 ± 146**	9.4 ± 1.9	101.2 ± 15.4	
Infasurf	278 ± 22	2,648 ± 137**	11.2 ± 2.8	94.9 ± 12.1	
Lipids alone	285 ± 26	4,062 ± 230*	11.4 ± 2.4	105.9 ± 5.5	
HCl-instilled rabbits (n = 8 per group)	
S-MB + SP-C33 UCLA		2,531 ± 176		83.9 ± 6.5	
S-MB		3,337 ± 228**		87.2 ± 5.4	
SP-C33 UCLA		3,203 ± 235**		98.3 ± 2.0	
Infasurf		3,874 ± 172**		103.9 ± 3.5***	
Lipids alone		4,623 ± 224*		105.9 ± 2.7***	
Notes.

* p < 0.01 vs. all other surfactant preparations.

** p < 0.05 vs. S-MB + SP-C33 UCLA surfactant.

*** p < 0.05 vs. S-MB + SP-C33 UCLA and S-MB surfactant.

Data analysis

Data are expressed as means ± standard error (SEM). Statistical differences were estimated using t-tests and analyses of variance (ANOVA). Student’s t-test was used for comparisons versus control values. Between groups comparisons at various time-points were done by one-way ANOVA and time courses were analyzed with one-way repeated measure ANOVA. A p value <0.05 was considered to indicate a significant difference.

Results

S-MB, SP-C33 UCLA and S-MB + SP-C33 UCLA surfactant and Infasurf (positive control) all had very high surface activity in captive bubble experiments and reached minimum surface tension values ≤ 1 mN/m during each of ten consecutive cycles of dynamic cycling (rate of 20 cycles/min, Fig. 1). Lipids alone (negative control) reached significantly higher minimum surface tension values of 16 mN/m (p < 0.001) than the one- and two peptide/lipid mixtures and Infasurf. Therefore, the relative order for the surfactant activities determined with captive bubble surfactometry for the various preparations were as follows: S-MB surfactant ∼ SP-C33 UCLA surfactant ∼ S-MB + SP-C33 UCLA surfactant ∼ Infasurf ≫ Lipids alone.

Figure 1 Surface activity of synthetic lung surfactants, clinical surfactant, and synthetic lipids only on the captive bubble surfactometer.

Minimum and maximum surface tension values are plotted for synthetic lipids with 3% (weight ratio) Super Mini-B (S-MB), 3% SP-C33 UCLA or 1.5% S-MB + 1.5% SP-C33 UCLA, clinical surfactant (Infasurf), and synthetic lipids alone. Synthetic lipids are 5:3:2 (weight ratio) DPPC:POPC:POPG. Surface activity of S-MB surfactant, Infasurf and synthetic lipids alone have been reported previously (Walther et al., 2005; Walther et al., 2010). Data are shown as mean ± SEM of n = 4.

In the surfactant-deficiency model, induced by repeated lung lavages in young adult rabbits, intratracheal instillation of surfactants with S-MB and/or SP-C33 UCLA and Infasurf quickly improved oxygenation and lung compliance, and continued to be biologically active until the end of the experimental period (Fig. 2). S-MB was more active than SP-C33 UCLA, but there was an additive effect of SP-C33 UCLA on S-MB function. The oxygenation and dynamic compliance curves obtained for Infasurf were close to those of S-MB surfactant. Instillation of “Lipids alone” had minimal effects on arterial oxygenation or compliance. The relative order of pulmonary activities for the various surfactant preparations in terms of both oxygenation and compliance was as follows: S-MB + SP-C33 UCLA surfactant > S-MB surfactant ∼ Infasurf > SP-C33 UCLA surfactant ≫ Lipids alone. The differences in oxygenation and compliance between S-MB + SP-C33 UCLA surfactant and S-MB, SP-C33 UCLA and Infasurf were statistically significant starting at 90 min after surfactant instillation, (p < 0.01) (Fig. 2).

Figure 2 Arterial oxygenation and dynamic compliance in surfactant-treated, ventilated rabbits with ARDS induced by repeated in vivo lavage.

Arterial partial pressure of oxygen (PaO2 in torr) and dynamic compliance (ml/kg/cm H2O) are shown as a function of time for the 5 groups of 7–8 ventilated rabbits treated with experimental surfactant at time 0, when PaO2 had dropped from >500 torr to <100 torr after standardized lung lavages. Rabbits were treated with synthetic lung surfactants (synthetic lipids + 3% Super Mini-B [S-MB], 3% SP-C33 UCLA or 1.5% S-MB + 1.5% SP-C33 UCLA) and clinical surfactant (Infasurf) as positive and synthetic lipids alone as negative control. Synthetic lipids are 5:3:2 (weight ratio) DPPC:POPC:POPG. From 90 min after surfactant treatment onwards, improvements in oxygenation and compliance for S-MB + SP-C33 UCLA surfactant differed significantly (p < 0.01) from the other surfactant preparations. Data are shown as mean ± SEM of groups of 7–8 rabbits.

In the surfactant-dysfunction model, induced by one or more intratracheal instillations of HCl in young adult rabbits, all peptide/lipid mixtures and Infasurf were less effective in improving oxygenation and lung compliance than in the lung lavage model (Figs. 2 and 3). Oxygenation and lung compliance continued to deteriorate after intratracheal instillation of synthetic lipids alone. Infasurf outperformed the peptide/lipid mixtures during the first 45 min after intratracheal instillation, but then started to lose activity as shown by deteriorating oxygenation and lung compliance and its surfactant activity was surpassed by the consistent performance of the peptide/lipid mixtures thereafter. The two-peptide surfactant mixture of S-MB and SP-C33 UCLA in DPPC:POPC:POPG lipids finally outperformed the one-peptide mixtures with SMB or SP-C33 UCLA and Infasurf, though they all succeeded in stabilizing and improving lung function after induction of chemical lung injury for the duration of the experiment. Instillation of synthetic lipids alone led to a continuous further deterioration of arterial oxygenation and lung compliance. The relative order of pulmonary activity in terms of both oxygenation and compliance (≥90 min) was given as: S-MB + SP-C33 UCLA surfactant > S-MB surfactant ∼ SP-C33 UCLA surfactants ∼ Infasurf ≫ Lipids alone. The differences in arterial oxygenation and lung compliance between the S-MB + SP-C33 UCLA surfactant and S-MB surfactant ∼ SP-C33 UCLA surfactants ∼ Infasurf were statistically significant starting at 90 min after surfactant instillation (p < 0.01) (Fig. 3).

Figure 3 Arterial oxygenation and dynamic compliance in surfactant-treated, ventilated rabbits with ALI induced by intratracheal instillation of 0.1 N hydrogen chloride (HCl).

Arterial partial pressure of oxygen (PaO2 in torr) and dynamic compliance (mL/kg/cm H2O) are shown as a function of time for the 5 groups of 8 ventilated rabbits treated with experimental surfactant at time 0, when PaO2 had dropped below 40% of the starting value after HCl instillation. Rabbits were treated with synthetic lung surfactants (synthetic lipids + 3% Super Mini-B [S-MB], 3% SP-C33 UCLA, or 1.5% S-MB + 1.5% SP-C33 UCLA) and clinical surfactant (Infasurf) as positive and synthetic lipids alone as negative control. Synthetic lipids are 5:3:2 (weight ratio) DPPC:POPC:POPG. From 90 min after surfactant treatment onwards, improvements in oxygenation and compliance for S-MB + SP-C33 UCLA surfactant differed significantly (p < 0.01) from the other surfactant preparations. Data are shown as mean ± SEM of groups of 8 rabbits.

Average (±SEM) protein content of BALF (Table 1) in the total group of lavaged animals (n = 39) increased almost 10-fold from 286 ± 10 µg/ml in the first 30 ml lavage to induce surfactant-deficiency to 2,776 ± 132 µg/ml in the first postmortem lavage. This increase was least for S-MB+SP-C33 UCLA and S-MB surfactant and highest for synthetic lipids alone. Protein content of BALF in the first postmortem lavage of the 40 HCl-instilled rabbits was higher than in the 39 lavaged rabbits (3,514 ± 143 vs. 2,776 ± 137 µg/ml, p < 0.001). Rabbits from the HCl group treated with S-MB + SP-C33 UCLA had the lowest and rabbits treated with lipids only had the highest protein values in the postmortem BALF samples among the 5 treatment groups (Table 1). Phospholipid values in BALF of lavaged rabbits also increased almost 10-fold from 10.5 ± 1.2 to 98.1 ± 10.6 µg/ml after treatment with each of the 4 surfactant preparations or lipid alone. Phospholipid content of the postmortem lavages of rabbits with HCl-induced ALI was comparable to that of the lavaged animals, although slightly, but statistically significantly, lower in rabbits treated with synthetic peptide surfactants than in Infasurf or lipids alone.

Discussion

The success of rescue therapy for chemical-induced ALI using synthetic surfactant with advanced and robust SP-B and SP-C mimics represents a significant breakthrough, as previous experiments have indicated only minimal or no improvement in lung function with animal-derived and first generation synthetic surfactant preparations (Lamm & Albert, 1990; Brackenbury et al., 2001; Zimmermann et al., 2010; Lampland et al., 2014). Lamm & Albert (1990) tested Survanta, a modified natural surfactant based on a bovine lung extract, in rabbit lungs after intratracheal injection of HCl, but found no improvement in arterial oxygenation. Brackenbury et al. (2001) treated rabbits pretreated with intratracheal HCl with natural ovine surfactant, bovine lipid extract surfactant and recombinant SP-C surfactant (Venticute) and found none of them effective. Recent studies by Zimmermann et al. (2010) and Lampland et al. (2014) compared intratracheal instillation and aerosol delivery of KL4 surfactant (Surfaxin®, Discovery Laboratories, Warrington, PA) with continuous positive airway pressure (CPAP) only in newborn pigs with acute lung injury after HCl instillation. Both KL4 preparations improved survival, but arterial oxygenation did not increase very much over the surfactant pretreatment values.

In a recent collaborative study with Tore Curstedt and Jan Johansson from the Karolinska Institute in Sweden, we examined the in vivo activities of Mini-B, i.e., a ‘truncated’ 34-residue predecessor of S-MB without the 7 amino-acid insertion sequence at the N-terminal (Waring et al., 2005), and/or SP-C33 with synthetic lipids in preterm newborn rabbits (Almlén et al., 2010). Treatment with either Mini-B or SP-C33 led to increased tidal and lung gas volumes, and combination treatment with these surfactant protein mimics demonstrated an additive effect in this validated animal model for neonatal respiratory distress. These results and the current data in lavaged and HCl-treated young adult rabbits indicate that synthetic surfactants containing analogs of both SP-B and SP-C may be superior to single-peptide surfactants in the treatment of RDS and ALI/ARDS.

The BALF findings indicate that treatment with an advanced two-peptide synthetic surfactant led to reduced protein values in the lavage fluid than those for single peptide synthetic surfactant or animal-derived surfactant with both SP-B and SP-C. This finding suggests that a more advanced and robust synthetic surfactant has the potential to diminish capillary-alveolar protein leakage and thereby reduce surfactant inhibition.

Animal models have their advantages and limitations. In this study animals were studied for 2 h after surfactant treatment, but longer duration of the experiments (at least 6 h) might better correlate with clinical outcome in ALI/ARDS patients. Lung lavage and HCl instillation each result in significant hypoxemia without hemodynamic effects that are fairly stable over at least the first 6–9 h (Rosenthal et al., 1998). In our hands mortality in the lung lavage model is less than the HCl-model, even though our primary goal of repeated lavages was to decrease arterial PO2 to values <100 torr and we accepted a higher limit (PaO2 <200 torr) after HCl instillation because values <80 torr were associated with a quick demise. In fact, the average PaO2 values after lung lavage were 53 torr (10% of the original values) and 117 torr (22% of original values) after HCl treatment. Lung compliance corresponded with oxygenation, lung lavage decreased lung compliance by 51% versus 39% after HCl treatment. These differences can be explained by a lesser degree of capillary-alveolar protein leakage after lung lavage than acid treatment that not only results in loss of active surfactant, but also leads to epithelial and endothelial damage, lung hemorrhages and copious edema formation resulting in severe surfactant dysfunction.

Conclusions

As opposed to animal-derived (Infasurf) or first generation synthetic (Surfaxin or Venticute) surfactant preparations, rescue therapy with a second generation synthetic (S-MB and SP-C33 UCLA in a synthetic three-lipid mixture) was highly effective in stabilizing and improving oxygenation and lung compliance in rabbits with chemical-induced lung injury.

Additional Information and Declarations

Competing Interests

Author Contributions

Animal Ethics

The authors declare there are no competing interests.

Frans J. Walther conceived and designed the experiments, performed the experiments, analyzed the data, contributed reagents/materials/analysis tools, wrote the paper, prepared figures and/or tables, reviewed drafts of the paper.

José M. Hernández-Juviel conceived and designed the experiments, performed the experiments, wrote the paper, prepared figures and/or tables, reviewed drafts of the paper.

Larry M. Gordon conceived and designed the experiments, analyzed the data, wrote the paper, reviewed drafts of the paper.

Alan J. Waring conceived and designed the experiments, performed the experiments, analyzed the data, contributed reagents/materials/analysis tools, wrote the paper, reviewed drafts of the paper.

The following information was supplied relating to ethical approvals (i.e., approving body and any reference numbers):

The animal studies were reviewed and approved by the Institutional Animal Care and Use Committee of the Los Angeles Biomedical Research Institute at Harbor-UCLA Medical Center: Research Project # 12507.

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
