# Peer review of "Synthetic surfactant containing SP-B and SP-C mimics is superior to single-peptide formulations in rabbits with chemical acute lung injury"

_PeerJ, doi:10.7717/peerj.393_

## Round 0.1 · original submission · Minor Revisions

Please respond to the minor comments of the reviewers.

Reviewer 1 ·

Basic reporting

No Comments.

Experimental design

Please clarify the temperature of the saline used to induce ARDS.

Validity of the findings

Fig 1,minimum surface tension. Most of the data in Fig 1 has been previously published. Please clarify the new data that has not published before.

Additional comments

Generally, the data are well presented using the standard model for ARDS. Additional study with longer periods(at least 6hr of study after surfactant treatment) would strength this paper.

Reviewer 2 ·

Basic reporting

This is a generally well-written article. There are some minor typographical errors that should be corrected. Specifically:

Abstract, line 4: "...absence of specific antidotes besides has...". Either the word "besides" should be deleted or the antidote the authors intended to indicate be included.

Abstract, line 17: "...Mechanism_s_ of action..."

Introduction, line 8: The sentence should end in a period, not a comma.

Experimental design

No comments.

Validity of the findings

It would be helpful to the reader in figures 2 and 3 and in the legends indicating the time-points where the results in improvements in oxygenation and compliance for S-MB+SP-C33 UCLA differ significantly from the other surfactant preparations.

Additional comments

A sentence clarifying or elaborating on exactly how SP-C33 differs from native SP-C (substituting leucines for valines for example) would be helpful to the reader.

A sentence in the discussion concerning the relatively short-duration of the experiment related to clinical outcomes in ARDS-ALI (not examined in this kind of study) would be worth including.

---

## Round 0.2 · accepted · Accept

Authors have improved the manuscript including the suggestions of the referees, therefore now it is in enough quality.